# Multiplayer Information Asymmetric Contextual Bandits

**William Chang**                                                          *chang314@g.ucla.edu*
*Department of Mathematics, UCLA, Los Angeles, CA, USA*

**Yuanhao Lu**                                                             *terrylu@princeton.edu*
*Princeton University, Princeton, NJ, USA*

**Reviewed on OpenReview:** *https://openreview.net/forum?id=nMCJ8bFq4B*

## Abstract

Single-player contextual bandits are a well-studied problem in reinforcement learning that has seen applications in various fields such as advertising, healthcare, and finance. In light of the recent work on *information asymmetric* bandits Chang et al. (2022); Chang and Lu (2023), we propose a novel multiplayer information asymmetric contextual bandit framework where there are multiple players each with their own set of actions. At every round, they observe the same context vectors and simultaneously take an action from their own set of actions, giving rise to a joint action. However, upon taking this action the players are subjected to information asymmetry in (1) actions and/or (2) rewards. We designed an algorithm `LinUCB` by modifying the classical single-player algorithm `LinUCB` in Chu et al. (2011) to achieve the optimal regret $O(\sqrt{T})$ when only one kind of asymmetry is present. We then propose a novel algorithm `ETC` that is built on explore-then-commit principles to achieve the same optimal regret when both types of asymmetry are present.

## 1 Introduction

The problem of Multi-armed Bandits (MAB) is one of the most well-studied classic reinforcement learning problems. The algorithms in the field are designed to find an optimal balance between the exploration-exploitation tradeoff dilemma. In the traditional setting of this problem, a single agent chooses one action (arm) from $m$ available actions over numerous iterations, where each action gives off a reward sampled from some unknown sub-Gaussian distribution. The primary objective is to minimize the agent's *regret*, defined as the difference between the expected reward of the agent's chosen actions and that of the optimal actions. Thus, the success of a policy can be measured by the *regret* as a function of time (number of actions taken). Under this classical setting, Lai and Robbins (1985) showed that no policy can achieve better than $O(\sqrt{T})$ regret. The UCB algorithm first attains this lower bound.

Although single-player MABs are well-studied, they fail to model more complex real-world problems involving multiple participants. Recently, there has been escalating interest in cooperative multiplayer MAB challenges, wherein several agents aim to maximize their aggregate expected returns collaboratively Chang and Lu (2023); Chang et al. (2022); Wang et al. (2020); Brânzei and Peres (2021); Pacchiano et al. (2023). Although these problem settings extend the MAB problems into multiple players, they still remain restrictive in real-world applications in these three aspects:

(1) These settings do not model the agents' access to information that might help agents predict the reward quality of an action (i.e. no context vectors).

(2) These settings assume the rewards obtained by each player are independent of the actions taken by other players (i.e. joint actions are not considered).

(3) These settings assume the agents can freely communicate their actions taken and rewards received to one another (i.e. information is perfectly symmetric).

To deal with restriction (1), prior works such as Chu et al. (2011) analyze the linear contextual bandit framework. Linear contextual bandits generalize the classical finite-armed MAB by allowing players to utilize side information to predict the quality of rewards. In each round of the contextual bandit problem, the agent observes one random context vector $\mathbf{x}$ per action, where the expectation of the reward distribution of that action is a zero-mean noise plus the inner product of the context vector $\mathbf{x}$ and an underlying parameter $\theta$ that is unknown to the players.[1] This framework Chu et al. (2011) relaxes the aforementioned restriction (1) by allowing agents to make use of the observed context $\theta$ to predict the rewards.

In this paper, we address restriction (2) by extending the contextual bandit framework into a cooperative multiplayer setting where the joint action of all players determines the reward distribution. Furthermore, we add novel *information asymmetry* to make our setting even more general. At each round, each player takes an action *individually* and *simultaneously* resulting in a joint action. This joint action generates the rewards for all players. In every round, all agents observe the same context vectors (one context vector per joint action). This multiplayer extension relaxes restriction (2).

To restrict communication between players and relax restrictions (3), we separately consider the following two types of information asymmetries: (1) *Action asymmetry* – At each round, each player receives the same reward but cannot observe other player's actions (the joint actions remains hidden to the players). (2) *Reward asymmetry* – at each round, each player receives an IID reward that can be only observed by themselves, while they are allowed to observe the actions of other players. Although players cannot communicate during the learning process, they are aware of the possible actions other players can take and can agree on a strategy beforehand.

**Our Contribution**   This is the first paper on multiplayer contextual bandits. We propose a multiplayer information asymmetric environment that was originally from the multi-armed bandit setting Chang et al. (2021); Chang and Lu (2023) and apply it to contextual bandits. We then propose two algorithms that are based on the single agent linear contextual bandit setting in Chu et al. (2011) called LinUCB. Remarkably, we show that by modifying LinUCB slightly, we obtain an algorithm that is able to take on both forms of information asymmetry. More specifically, through a coordination scheme, we are able to recover the same regret bound $O(\sqrt{T})$ as in the single-agent setting when the players receive the same reward but can't observe the other player's actions (Problem A). On the other hand, when the players receive their own IID reward but can observe the other player's actions (Problem B), we obtain the first sublinear regret bound of $O(\sqrt{T})$. Finally, when there are both types of information asymmetry (Problem C), we propose a new algorithm that involves principles in the classical Explore and then commit algorithm that achieves the same order regret bound.

**Related Works**   The single-player contextual bandit with linear payoff functions is a well-studied problem with efficient algorithmic solutions Agarwal et al. (2014); Agrawal and Goyal (2013). There are many variants to the single-player linear contextual bandit setting such as Agrawal and Devanur (2016) and Badanidiyuru et al. (2014) which consider bandits with constraints on resource allocations. Furthermore, Bouneffouf et al. (2017) studies the problem with restricted context vectors, Allesiardo et al. (2014) analyzes contextual bandits that do not need a hypothesis on stationary properties of contexts and rewards.

Linear contextual bandits have numerous real-world applications, encompassing healthcare, recommender systems, information retrieval, and risk management. For example, Durand et al. (2018) employs the contextual bandit framework to adaptively treat mice in the early stages of cancer. Li et al. (2010) and Bouneffouf et al. (2012) leverage contextual information to enhance mobile and news article recommendation systems. Bouneffouf et al. (2013) applies contextual bandits to optimize context-based information retrieval. Furthermore, Soemers et al. (2018) utilizes contextual bandits to adaptively distinguish between fraud and concept drifts in credit card transactions. Within machine learning, Laroche and Féraud (2017) employs contextual bandits for algorithmic selection in off-policy reinforcement learning, while Bouneffouf et al. (2014) integrates them to improve active learning.

---

[1]$\theta$ is global and independent of the actions. Moreover, $\theta$ is inherent to the contextual environment and does not change in between rounds.

We will now overview the literature on multiplayer bandits. Within the domain of multiplayer stochastic bandits, numerous studies permit restricted communication, as observed in prior research such as Martínez-Rubio et al. (2018; 2019); Szorenyi et al. (2013); Karpov et al. (2020); Tao et al. (2019). Recent studies, building upon the foundations laid by Chang et al. (2021), have delved into investigations of information asymmetry in the context of multiplayer bandits, as explored in works such as Mao et al. (2022); Kao et al. (2022); Mao et al. (2021); Kao (2022).

In cooperative multiplayer bandits, the objective is to determine the optimal arm from a set of shared arms among players. The communication structure between players is represented by a graph. This concept was first introduced by Awerbuch and Kleinberg (2008). Since then, several strategies have been proposed, including $\epsilon$-greedy Szorenyi et al. (2013), gossip UCB Landgren et al. (2016), accelerated gossip UCB Martínez-Rubio et al. (2019), and leader-based approaches Wang et al. (2020). The problem has also been explored in an adversarial setting by Bar-On and Mansour (2019), who introduced a strategy where followers adopt the EXP3 algorithm. Another line of research allows players to observe the rewards of their neighbors at each time step, based on their relative positions in the graph Cesa-Bianchi et al. (2016). Additionally, some studies have considered asynchronous settings where only a subset of players is active in each round Bonnefoi et al. (2017); Cesa-Bianchi et al. (2020). In the collision setting, when multiple players select the same arm, a collision occurs, preventing them from collecting rewards. This setting does not account for joint arms. An extension to the Lipschitz setting was explored in Proutiere and Wang (2019), where they introduced DPE (Decentralized Parsimonious Exploration), an algorithm designed to minimize communication while enabling players to maximize their cumulative rewards.

The concept of competing bandits was introduced by Liu et al. (2020). This model resembles the collision setting but incorporates player preferences. When multiple players select the same arm, only the highest-ranked player receives the reward. In this framework, a centralized CUB algorithm was proposed, where players communicate their UCB indices to a central agent. Cen and Shah (2022) demonstrated that logarithmic optimal regret can be achieved if the platform also manages transfers between players and arms. Jagadeesan et al. (2021) further explored this by considering a stronger equilibrium notion, where agents negotiate these transfers. Liu et al. (2020) also introduced an ETC algorithm that attains logarithmic optimal regret without requiring transfers, but assuming knowledge of the reward gaps. Sankararaman et al. (2021) extended this approach by eliminating the need for such knowledge. Furthermore, Liu et al. (2021) proposed a decentralized UCB algorithm incorporating a collision avoidance mechanism.

## 2 Preliminary

We consider information asymmetric contextual bandits, which is a generalization of the single player setting given in Chu et al. (2011). In particular, they propose a UCB-index based algorithm LinUCB, and we propose a multiplayer version with joint arms of this algorithm.

In particular, we suppose there are $m$ players, and each player $i$ can pick from a set of arms $\mathcal{A}_i$. For simplicity, we can assume that $|\mathcal{A}_i| = K$ although the case where each player has a different number of arms is easily generalizable (the coordination techniques still work as long as all the players know how many actions they have access to prior to learning). At every round $t$, each player will pick an arm from their action set without communication. This gives rise to a *joint* arm (which can be represented as a vector of actions from each player) $\mathcal{A} := \mathcal{A}_1 \times \cdots \times \mathcal{A}_m$ and can be denoted as $\boldsymbol{a}_t$ which produces a stochastic reward $r_{t,\boldsymbol{a}_t}$. We will use **bold** to denote any quantity that is a vector. Given a joint action $\boldsymbol{a}$ we define the term *corresponding action* for player $i$ to be the $i$th component in the vector $\boldsymbol{a}$. We shall use $T$ to denote the total number of rounds in the learning process. The collective goal of all the players is to maximize the total expected rewards up to horizon $T$.

Furthermore, at the start of each round, every player is given the same $K^m$ *context* vectors $\boldsymbol{x}_{t,\boldsymbol{a}} \in \mathbb{R}^d$ corresponding to each joint arm $\boldsymbol{a} \in \mathcal{A}$. Suppose that each contextual vector $\boldsymbol{x}_{t,\boldsymbol{a}}$ satisfies $\|\boldsymbol{x}_{t,\boldsymbol{a}}\|_{\ell_2} \leq L$ under the $\ell_2$ norm. The reward that is produced from pulling joint arm $\boldsymbol{a}$ satisfies the linear realizability assumption, that is,

$$\mathbb{E}[r_{t,\boldsymbol{a}}|\boldsymbol{x}_{t,\boldsymbol{a}}] = \langle \boldsymbol{x}_{t,\boldsymbol{a}}, \boldsymbol{\theta}^* \rangle \tag{1}$$

for some $\boldsymbol{\theta}^* \in \mathbb{R}^d$. This means that in order to determine which arms have the best context it is desirable to have an accurate estimate of $\boldsymbol{\theta}^*$.

Let $\boldsymbol{a}_t$ be the joint arm that is selected at round $t$. Furthermore, let $\boldsymbol{a}_t^*$ be the best arm at round $t$. That is $\boldsymbol{a}_t^* = \arg\max_{\boldsymbol{a} \in \mathcal{A}} \langle \boldsymbol{x}_{t,\boldsymbol{a}}, \boldsymbol{\theta}^* \rangle$. To understand the success of a policy, we shall use the notion of regret $R_T$ up to horizon $T$, defined as

$$R_T = \sum_{t=1}^{T} \langle \boldsymbol{x}_t^*, \boldsymbol{\theta}^* \rangle - \langle \boldsymbol{x}_{t,\boldsymbol{a}_t}, \boldsymbol{\theta}^* \rangle = \sum_{t=1}^{T} \langle \boldsymbol{x}_t^* - \boldsymbol{x}_{t,\boldsymbol{a}_t}, \boldsymbol{\theta}^* \rangle \qquad (2)$$

In Chu et al. (2011), they were able to prove that LinUCB attains $O(\sqrt{T})$ regret, which matches the lower bound for this problem. We can use the lower bound from the single agent setting but on the $K^M$ joint actions. We now state the information asymmetric problems we will be studying taken from Chang et al. (2022); Chang and Lu (2023). They are as follows (recall that all players receive the same contexts for all the joint actions each round).

**Problem A:** Information asymmetry in actions. At every round, after a joint action $\boldsymbol{a}$ is taken, the agents cannot observe the actions of the other players but all players receive the same rewards.

**Problem B:** Information asymmetry in rewards. At every round, after a joint action $\boldsymbol{a}$ is taken, agents only observe their own i.i.d. copy of the reward but they can observe the actions of other players.

**Problem C:** Information asymmetry in both actions and rewards. This combines the challenges in problem A and problem B where every round the players get their own i.i.d. reward (without seeing other players' rewards) *and* they cannot observe the actions taken by other players.

To be precise, in problems B and C, since each player obtains a different reward, we should use $R_T^i$ to be the regret for player $i$. However as the distributions of the rewards have the same mean, and regret is defined under expectation, it follows that even in this setting each player experiences the same regret.

## 2.1 Challenges in the Contextual Bandit Setting

In this section we compare our work to that given in Chang et al. (2022). In their paper, they study the information asymmetry bandit problem for the classical multi-armed bandit setting. For problem $A$, information asymmetry in only actions, all the players receive the same reward feedback but are unable to communicate as well as observe the other player's actions at each round. However, because they receive the same reward feedback, if they are to correctly infer the other player's actions then they are able to maintain the same UCB estimates of all the arms. Similarly, in the contextual bandit setting, they observe the same rewards as well as the same contexts. Thus, they are able to maintain the same estimate for $\boldsymbol{\theta}^*$ as well as the same confidence set. The novelty is constructing a way to break ties when two arms have the same LinUCB index so that each player can accurately infer the correct action that is taken at the time step despite not being able to observe the actions of the other players. This is where Definition 1 plays a role in Algorithm 1.

On the other hand, problem B, which is information asymmetry in rewards is a bit more challenging. Since each player observes only their own IID copy of their reward, they will maintain different estimates of $\boldsymbol{\theta}^*$ (and therefore have different confidence sets for this parameter as well). In the bandit's case studied in Chang and Lu (2023), this issue was addressed using a UCB-interval algorithm, where initially all the arms were pulled in a predefined order. In that paper, each player maintains for each arm their own UCB-interval, and when two UCB intervals are disjoint the suboptimal arm gets eliminated. However, such an elimination method no longer applies to the contextual bandit case because at every round the distribution of each arm changes in accordance with the context it receives. However, we make this problem easier by assuming the context vectors are stochastically generated. This makes it easier for players to coordinate their actions by simply improving their estimates of $\boldsymbol{\theta}^*$, which can be done by pulling any joint arm. In comparison to the standard MAB, the empirical mean of the joint action is only improved when that action is taken.

## 3 Main Results

### 3.1 LinUCB

We describe how LinUCB works from Chu et al. (2011) using the multiagent environment. In particular, this algorithm maintains an estimate of $\boldsymbol{\theta^*}$ by solving the following least squares estimator (for player $i$).

$$\boldsymbol{\theta}_t^i = \arg\min_{\boldsymbol{\theta} \in \mathbb{R}^d} \left( \sum_{t=1}^T (r_{t,\boldsymbol{a}}^i - \langle \boldsymbol{\theta}, \boldsymbol{x}_{t,\boldsymbol{a}_t} \rangle)^2 + \lambda \|\boldsymbol{\theta}\|_{\ell_2}^2 \right) \tag{3}$$

which has solution

$$\boldsymbol{\theta}_t^i = V_t^{-1} \sum_{t=1}^T \boldsymbol{x}_{t,\boldsymbol{a}} r_{t,\boldsymbol{a}} \tag{4}$$

where $V_t$ are $d \times d$ matrices

$$V_0 = \lambda I \text{ and } V_t = V_0 + \sum_{t=1}^T \boldsymbol{x}_{t,\boldsymbol{a}_t} \boldsymbol{x}_{t,\boldsymbol{a}_t}^\top \tag{5}$$

This $\boldsymbol{\theta}_t^i$ gives an estimate of $\theta^*$ in the contextual bandit setting.

For the estimate of $\boldsymbol{\theta^*}$, we construct a confidence interval $C_t(\theta)$ which is the set of vectors in $\mathbb{R}^d$ that are at most a certain distance away from $\theta$ under the norm $\|\boldsymbol{v}\|_{V_{t-1}}^2 = \boldsymbol{v}^\top V_{t-1} \boldsymbol{v}$. More explicitly our confidence set is,

$$C_t(\boldsymbol{\theta}) = \{\boldsymbol{v} \in \mathbb{R}^d : \|\boldsymbol{v} - \boldsymbol{\theta}\|_{V_{t-1}}^2 \leq \beta_T\} \tag{6}$$

For each arm, each player $i$ can construct an Upper Confidence Bound by solving the following optimization problem

$$\max_{\boldsymbol{\theta} \in C_t(\theta_t^i)} \langle \boldsymbol{\theta}, \boldsymbol{x}_{t,\boldsymbol{a}} \rangle \tag{7}$$

This optimization problem has the solution

$$\langle \boldsymbol{x}_{t,\boldsymbol{a}}, \boldsymbol{\theta_t^i} \rangle + \sqrt{\beta_t} \|\boldsymbol{x}_{t,\boldsymbol{a}}\|_{V_{t-1}^{-1}} \tag{8}$$

and each player will pick the arm with the highest index.

In the classical case, $\beta_t$ can be chosen as

$$\sqrt{\beta_t} = \sqrt{\lambda} m_2 + \sqrt{2 \log\left(\frac{1}{\delta}\right) + d \log\left(\frac{d\lambda + tL^2}{d\lambda}\right)} \tag{9}$$

where $\lambda$ is used to initialize $V_0$ and is in this setting can be any positive number.

In the following subsections, we will generalize the LinUCB algorithm from Chu et al. (2011) to account for the information asymmetries namely action asymmetry (Problem A) and reward asymmetry (Problem B), and state their regret bounds.

## 3.2 Asymmetry in Actions

---

**Algorithm 1** `LinUCB-A` for asymmetry in actions

---

1: **Input:** $\alpha > 0$, $K, m, d \in \mathbb{N}$
2: $V_t \leftarrow I_d$,
3: $\boldsymbol{b} \leftarrow 0_d$
4: **for** $t = 1, 2, 3, \ldots, T$ **do**
5: $\quad \theta_t \leftarrow V^{-1}\boldsymbol{b}$
6: $\quad$ Observe $K^m$ arm contexts $\boldsymbol{x}_{t,\boldsymbol{a}}$ for each joint arm $\boldsymbol{a} \in \mathcal{A}$.
7: $\quad$ **for** each joint arm $\boldsymbol{a} \in \mathcal{A}$ **do**
8: $\qquad p_{t,\boldsymbol{a}} \leftarrow \theta_t^\top \boldsymbol{x}_{t,\boldsymbol{a}} + \alpha\sqrt{\boldsymbol{x}_{t,\boldsymbol{a}}^\top V^{-1}\boldsymbol{x}_{t,\boldsymbol{a}}}$
9: $\quad$ **end for**
10: $\quad$ All players take their corresponding action for $\boldsymbol{a}_t \in \arg\max_{\boldsymbol{a}} p_{t,\boldsymbol{a}}$, where joint action $\boldsymbol{a}_t$ is chosen so that it's smallest by Definition 1. [2]
11: $\quad$ Observe reward $r_t \in \{0, 1\}$
12: $\quad$ Update $V \leftarrow V + \boldsymbol{x}_{t,\boldsymbol{a}_t}\boldsymbol{x}_{t,\boldsymbol{a}_t}^\top$.
13: $\quad$ Update $\boldsymbol{b} \leftarrow \boldsymbol{b} + \boldsymbol{x}_{t,\boldsymbol{a}_t}r_t$.
14: **end for**

---

In this section, we generalize the LinUCB algorithm action asymmetry (Problem A) and call it `LinUCB-A`. This is the setting where each player receives the same reward but is unable to observe the other player's actions at every round. Since the feedback from all the players is the same, the only challenge comes in inferring the other player's actions. In particular, when two joint actions have the same UCB index, there needs to be a way to break ties. Therefore, we define the following ordering on the joint arm space.

**Definition 1.** *Number the players $1, \ldots, m$ and the $K$ individual actions, and consider each set of joint action $\boldsymbol{a}$ as an $m$ digit number with each digit corresponding to the joint action. Call this base $K$ number $N_{\boldsymbol{a}}$. For joint action $\boldsymbol{a}, \boldsymbol{b} \in \mathcal{A}$, we say that $\boldsymbol{a} < \boldsymbol{b}$ if $N_{\boldsymbol{a}} < N_{\boldsymbol{b}}$.*

This is similar to what is done in Chang et al. (2022). The idea is that even though the players cannot observe, the other player's actions, because they obtain the same feedback, they can infer what the other players are doing as long as they have a way to break ties should two joint actions have the same index. Because of this coordination, the players are behaving as if they were single agent in a larger joint action space. From this, we can deduce the following regret bound.

**Theorem 2.** *In the action asymmetric (Problem A) contextual bandit setting where the context vectors, the frequentist regret bound of Algorithm 1 is*

$$R_T = Cd\sqrt{T}\log(TL) \tag{10}$$

*Proof.* See Corollary 19.3 of Lattimore and Szepesvári (2020). $\qquad\square$

We note that this bound truly reduces to the single agent setting case as it doesn't even grow with the number of arms. This is because the success of the algorithm only depends on the accuracy in the estimate of $\boldsymbol{\theta^*}$. In comparison, in the multiarmed bandit problem, the regret grows with action space because every arm needs to be estimated.

## 3.3 Asymmetry in Rewards

In this section, we generalize the LinUCB algorithm reward asymmetry (Problem B) and call it `LinUCB-B`. This is the setting where each player receives an i.i.d copy of the reward but is able to observe the other player's actions at every round. This algorithm is similar to `LinUCB-A` but takes into account that the reward feedback is different for different players.

---
**Algorithm 2** `LinUCB-B` for asymmetry in rewards

---
1: **Input:** $\alpha > 0$, $K, m, d \in \mathbb{N}$ $\boldsymbol{a}_t \leftarrow \lambda I_d$, where $\lambda = T^{\frac{1}{2}}$ $\boldsymbol{b}^i \leftarrow 0_d$
2: **for** $t = 1, 2, 3, \ldots, T$ **do**
3:     Each player $i$ updates $\boldsymbol{\theta}_t^i \leftarrow V^{-1} \boldsymbol{b}^i$
4:     Each player Observe $K^m$ arm contexts $\boldsymbol{x}_{t,\boldsymbol{a}}$ for each joint arm $\boldsymbol{a} \in \mathcal{A}$.
5:     **for** each joint arm $\boldsymbol{a} \in \mathcal{A}$ **do**
6:         Each player $i$ updates $p_{t,\boldsymbol{a}}^i \leftarrow (\boldsymbol{\theta}_t^i)^\top \boldsymbol{x}_{t,\boldsymbol{a}} + \alpha \sqrt{\boldsymbol{x}_{t,\boldsymbol{a}}^\top V^{-1} \boldsymbol{x}_{t,\boldsymbol{a}}}$
7:     **end for**
8:     Each player $i$ chooses their corresponding action for their observed $\boldsymbol{a}_t = \arg\max_{\boldsymbol{a}} p_{t,a}^i$.
9:     Each player observes the other player's actions.
10:     Each players observes an I.I.D. reward $r_t^i \in \{0, 1\}$
11:     Each player updates $V = V + \boldsymbol{x}_{t,\boldsymbol{a}_t} \boldsymbol{x}_{t,\boldsymbol{a}_t}^\top$
12:     Each player $i$ updates $\boldsymbol{b}^i \leftarrow \boldsymbol{b}^i + \boldsymbol{x}_{t,\boldsymbol{a}_t} r_t^i$
13: **end for**

---

The central idea is to modify $\lambda$ and $\sqrt{\beta_t}$ so that each player's confidence set is small enough so that for some distribution of context vectors there is a very high probability that all the players agree on the optimal arm for each particular round. In doing so, we allow the players to implicitly coordinate their actions without any need for the players to communicate during the learning process. More specifically we set,

$$\sqrt{\beta_T} = O\left(T^{c/2} \|\boldsymbol{\theta}^*\|_2\right) \tag{11}$$

with $c = \frac{1}{2}$ and our initialization for $V_0 = \lambda I$ is

$$\lambda = T^c. \tag{12}$$

Compare this to equation 9. In particular, the ratio of $\beta_T / \lambda$ is much smaller for this setting than it is for the setting in problem A. This is because $\frac{\beta_T}{\lambda}$ (as we show in Lemma 7) is the lower bound for the radius of the confidence interval for our estimate of $\boldsymbol{\theta}^*$, and we need these to be sufficiently small in order for the aforementioned coordination to occur.

We shall show that remarkably, even using the same algorithm as Algorithm 1 (with just modifying $\lambda = \sqrt{T}$) we can obtain a regret bound that is still sublinear. Note in this algorithm that the rewards $r_t^i$ at time $t$ are indexed by $i$, since each player observes their own copy of an IID reward, without seeing the other players' copy. Therefore each player has their own estimate of $\boldsymbol{b}^i$ as well. This also causes their estimate of the parameter $\boldsymbol{\theta}^*$ to be different from each other, resulting in different confidence sets for $\boldsymbol{\theta}^*$ as well. The explicit algorithm is stated in Algorithm 2, where the quantities that are now different for each player have a superscript $i$ attached to them. For this algorithm to work we have to assume that the context vectors are generated by some fixed (but unknown) distribution in the unit ball of radius $L$.

To see that it is impossible to obtain sublinear regret using adversarial contexts consider the 2 player environment, and suppose each player has two arms $\{1, 2\}$. Then we consider the following $2 \times 2$ matrix where the row labels are the actions of one player and the column labels are the actions for the other player. Furthermore, each entry corresponds to a context vector for that joint action.

$$\begin{array}{c} \quad\quad 1 \quad\quad 2 \\ \begin{array}{c} 1 \\ 2 \end{array} \left[ \begin{array}{cc} \boldsymbol{v}_t & \boldsymbol{0} \\ \boldsymbol{0} & \boldsymbol{v}_t' \end{array} \right] \end{array}$$

where either $\boldsymbol{v}$ is the best contex vector, and $\boldsymbol{v} \sim \boldsymbol{v}'$ in that $\langle \boldsymbol{v}, \boldsymbol{\theta}^* \rangle$ and $\langle \boldsymbol{v}', \boldsymbol{\theta}^* \rangle$ are really close to each other. When they are sufficiently close since the players have IID rewards, their estimates $\boldsymbol{\theta}_t^i$ will also be slightly different. If two players disagree on which context vector is the best, they will obtain 0 reward. For the appropriate context vectors, this happens with constant probability, and thus we obtain constant

regret. Note that we refrained from setting $\boldsymbol{v} = \boldsymbol{v}'$ because when two context vectors are the same the players can still coordinate by ordering the arms as in Definition 1 which was done while studying Problem A. Furthermore, this is not an issue that shows up in the single-agent setting because even if the player is unable to decide which of $\boldsymbol{v}$ or $\boldsymbol{v}'$ is better, it doesn't matter because pulling either incurs little regret. In the multiplayer settings, the issues show up when two players *disagree* on which arm to select for many of the rounds.

Therefore, let $\psi(x)$ be a Lebesgue integrable probability distribution density of this ball that contains the context vectors and suppose that $\|\psi\|_{L^\infty} < \infty$. Note that it does not need to be continuous. Letting $\mu$ be the Lebesgue measure over $(\mathbb{R}^d, \mathcal{M})$ (with $\mathcal{M}$ is the $\sigma$-algebra of Borel sets), it follows that for any subset $U$, we have

$$\mathbb{P}_\mu(x \in U) = \int_U \psi(x) d\mu(x) < \|\psi\|_{L^\infty} \mu(U)$$

It follows that as $\mu(U) \to 0$, we have $\mathbb{P}_\mu(x \in U) \to 0$ as well. This also means that as the players refine their estimate of $\theta^*$, the chances that the players will disagree on which arm to pull will decrease in probability. This intuition is formalized in Lemma 7 and Lemma 8.

We can now state the regret bound of Algorithm 2 under reward asymmetry (Problem B).

**Theorem 3.** *In the reward asymmetric (Problem B) contextual bandit setting where the context vectors are distributed with fixed distribution the frequentist regret bound of Algorithm 2 is,*

$$R_T = O(mK^{2m}L^d\sqrt{T}\log(T)) \tag{13}$$

The proof of this is given in the supplementary materials.

Note that this result depends on the number of actions. That's because in order for the players to be coordinated the context vectors of the joint action to have be sufficiently far. This is formalized in Lemma 7

### 3.4 Asymmetry in Both Rewards and Actions

In this section, we propose `ETC` which will be applied to Problem C. In the previous section, we showed that LinUCB does well even when the rewards are IID (problem B). This is because in this setting the players are still able to observe the other player's actions and therefore they can make the correct updates. However, in this setting, as they cannot observe the other player's actions, we cannot guarantee each player will attempt to pull the same joint arm. In particular, at the beginning of the learning process, when the estimate of $\boldsymbol{\theta}$ isn't very accurate for any player, this increases the probability of mis-coordination.

We circumvent this by giving an exploration sequence of time $T^\alpha$ where it will be shown in the proof that $\alpha = \frac{1}{2}$ is optimal. During the exploration sequence, all the players will pull arm **1** (or any other fixed arm(s)) as long as they agree on which ones to pull at each round. In this time they will update their $\boldsymbol{V_t}$ and $\boldsymbol{b_i}$ parameters. After the exploration phase they will run regular Lin UCB, but *they will not update their $V_t$ and $b_i$ values.* The idea is that after sufficient exploration they will each have (different) but accurate estimates of $\boldsymbol{\theta}$. Since the context vectors are generated at random (rather than adversarial), there is a high probability that they will be able to successfully coordinate pulling the best action at every round.

Similar to Algorithm 2, our choice of $\lambda = \sqrt{T}$ is important. By selecting a large enough $\lambda$ we ensure that the confidence ball for $\boldsymbol{\theta}$ is sufficiently small. However, we cannot choose $\lambda$ too big, or else our confidence ball for $\boldsymbol{\theta}$ will not contain $\boldsymbol{\theta}$ with sufficiently high probability.

We have the following regret bound for this algorithm

**Theorem 4.** *In the reward and action asymmetric (problem C) contextual bandit setting where the context vectors are distributed with fixed distribution the frequentist regret bound of the algorithm is*

$$R_T = O(mK^{2m}L^d\sqrt{T}\log(T)) \tag{14}$$

The proof of this is given in the supplementary materials.

---

**Algorithm 3** `ETC` for asymmetry in rewards and actions

---

1: **Input:** $\beta_T > 0$, $K, m, d \in \mathbb{N}$, exploration parameter $T^\alpha$.
2: $\boldsymbol{a}_t \leftarrow \lambda I_d$, with $\lambda = T^\alpha$ where $\alpha = \frac{1}{2}$
3: $b_t^i \leftarrow 0_d$
4: **for** $t = 1, 2, 3, \ldots, T^\alpha$ **do**
5:     All players will pull the corresponding arm to the joint action **1**.
6:     Update $V_{t+1} \leftarrow V_t + \boldsymbol{x}_{t,\boldsymbol{a}_t}\boldsymbol{x}_{t,\boldsymbol{a}_t}^\top$
7:     Update $b^i \leftarrow b^i + \boldsymbol{x}_{t,\boldsymbol{a}_t}r_t^i$
8: **end for**
9: $\boldsymbol{\theta}_t^i \leftarrow V^{-1}b^i$
10: **for** $t = T^\alpha + 1, ..., T$ **do**
11:     Observe $K^m$ arm contexts $\boldsymbol{x}_{t,\boldsymbol{a}}$ for each joint arm $\boldsymbol{a} \in \mathcal{A}$.
12:     **for** each joint arm $\boldsymbol{a} \in \mathcal{A}$ **do**
13:         $p_{t,\boldsymbol{a}}^i \leftarrow (\boldsymbol{\theta}_t^i)^\top \boldsymbol{x}_{t,\boldsymbol{a}} + \sqrt{\beta_T}\sqrt{\boldsymbol{x}_{t,\boldsymbol{a}}^\top V^{-1}\boldsymbol{x}_{t,\boldsymbol{a}}}$
14:     **end for**
15:     Each player chooses their corresponding action for their observed $\boldsymbol{a}_t = \arg\max_{\boldsymbol{a}} p_{t,a}$.
16: **end for**

---

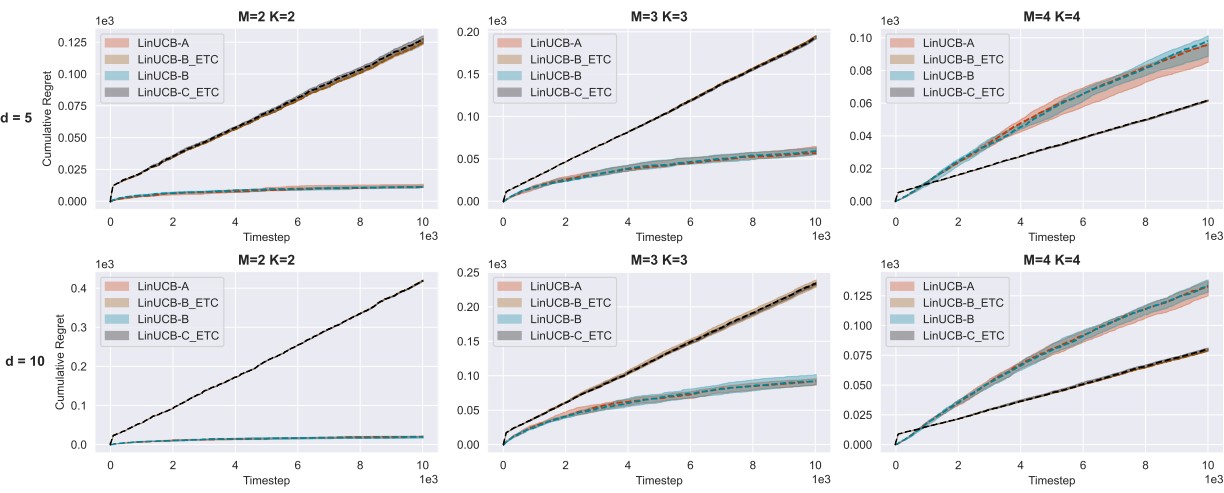

Figure 1: Regret plots comparing different algorithms to different information asymmetry. Red is the regret of `LinUCB-A` on Problem A (asymmetry in actions). Brown is the regret for `ETC` on Problem B (asymmetry in rewards). Green is the regret plot for `LinUCB-B` on Problem B. Finally black is the regret for `ETC` on Problem C (asymmetry in both rewards and actions).

Similar to the regret bound of Algorithm 2 provided by Theorem 3, this depends on the number of actions due to the fact that every round the players are miscoordinated (i.e. when the context vectors are too close to each other), we incur linear regret.

## 4 Experiments

In this section, we execute simulations to corroborate the empirical efficacy of the proposed algorithms in this paper. In Figure 1, we plot the regret versus time for both algorithms `LinUCB-A` and `LinUCB-B`. It should be emphasized these algorithms assume different types of asymmetry: `LinUCB-A` assumes action asymmetry while `LinUCB-B` assumes reward asymmetry.

### 4.1 Experiment Details

We conduct the simulations using $\boldsymbol{\theta}$ and context vectors $x$ uniformly sampled from the unit cube $[0, \frac{1}{\sqrt{d}}]$. This parametrization ensures that $\|\boldsymbol{\theta}\|_{\ell_2}$ and $\|\boldsymbol{x}\|_{\ell_2}$, measured using the $\ell_2$ norm, does not exceed $L = \sqrt{d}$, in line with the constraints of our problem setting. Furthermore, it's clear this uniform distribution is bounded over our space for $x$. Each reward is set to be Gaussian, and the standard deviation of them is randomly uniformly pre-selected to be from the range $[0, 1]$. For each environment, the simulations were executed over $T = 10,000$ rounds. We repeat these simulations 5 times to compute the median regret and report the 95% confidence interval. The hyperparameter $\beta_T$ is set to $\sqrt{T}$ for all algorithms analyzed.

In the proceeding section, we perform the experiment on environments with $m$ and $K$ equal to $2, 3, 4$ respectively, with $d = 5, 10$. Moreover, we use `LinUCB-B_ETC` to denote the `ETC` algorithm run on problem B. Similarly, `LinUCB-C_ETC` is used to denote the `ETC` algorithm run on problem C.[3]

### 4.2 Analysis

We note that since `LinUCB-A` is the same algorithm as the single-player setting but with an added ordering, it serves as the baseline to compare with our other algorithms. `LinUCB` performs relatively well as compared to `LinUCB-A` but `LinUCB-A` tends to perform better. This is because while `LinUCB-A` has the more favorable feedback, `LinUCB-B` has a larger $\lambda$ parameter which encourages less exploration. In the analysis, this affects the probability of the "good event" that the $\boldsymbol{\theta}$ will stay within the confidence ball. However, in our simulations, due to the small environment, it's unlikely that the 'bad' events will occur. Therefore, in this case, it's more favorable to do less exploration.

We note that `ETC` appears to be piecewise linear. In particular, the first piece which only occurs for $\sqrt{T}$ rounds is steeper as this is the exploration phase. In the second piece, the algorithm takes the parameters taken from the periods of exploration and then runs `LinUCB` without updating these parameters. Philosophically, the slope of the regret curve reflects an algorithm's learning. Because the parameters don't update, `ETC` does not perform better as the rounds continue (which is different than the standard `LinUCB`, the slope of the regret curve remains constant. Despite being piecewise linear, however, asymptotically the regret will still grow in the same order as `LinUCB-B`.

In comparing `ETC` and `LinUCB-B` on the asymmetry in the rewards environment (Problem B), we note that, `LinUCB-B` performs superior. However, `ETC` is more robust as it achieves around the same level of performance in both Problem B And Problem C settings. This makes sense because `ETC` is a fully coordinated algorithm so it does not need to rely on observing actions to achieve its performance.

## 5 Conclusions and Future Work

In this paper, we adapted LinUCB from Chu et al. (2011) to the multiagent setting with different types of information asymmetry. Namely, we studied action asymmetry (Problem A) where each player receives the same reward but cannot observe other player's actions. Using a coordination scheme we were able to reduce this to the single agent setting and obtain an $O(\sqrt{T})$ regret bound. On the other hand, we also studied reward asymmetry (Problem B) where each player receives an iid copy of the reward but can observe the other player's actions. In this setting, we can prove that if the context vectors are distributed with a fixed distribution (rather than adversarial), then we obtain a $O(\sqrt{T})$ regret bound. We were able to achieve this using the same algorithm as that in Problem A but modifying $\lambda$ to be $\sqrt{T}$. Both of these regret bounds are the first for this setting. For asymmetry in both (Problem C), we proposed a fully coordinated `ETC` algorithm which did exploration for the first $\sqrt{T}$ rounds and then ran `LinUCB` for the remaining time, which achieved the same regret as the multiplayer `LinUCB`. Finally, we corroborated our results with some simulations.

In our regret bound, we have a dependence on $K^M$, the number of joint actions. This occurs because we assumed the contextual vectors are randomly generated and so we need these to be "well behaved" enough so

---

that the players can coordinate. However, this takes away from the true power of linear reward models where the regret bound doesn't depend on the number of actions. For future work, we can perhaps show that this is necessary via a lower bound, or propose a new setting (perhaps the players pull their arms successively rather than simultaneously).

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

# 6 Supplementary Material

## 6.1 Concentration Lemmas

The following is taken from Theorem 20.5 of Lattimore and Szepesvári (2020). It gives us the size of the ball that contains $\boldsymbol{\theta}$ with high probability.

**Lemma 5.** *Let $\delta \in (0,1)$. Then, with probability at least $1 - \delta$, it holds that for all $t \in \mathbb{N}$,*

$$\left\| \hat{\theta}_t - \theta_* \right\|_{V_t(\lambda)} < \sqrt{\lambda} \left\| \theta_* \right\|_2 + \sqrt{2 \log \left( \frac{1}{\delta} \right) + \log \left( \frac{\det V_t(\lambda)}{\lambda^d} \right)}.$$

*Furthermore, if $\left\| \theta_* \right\|_2 \leq L$, then $\mathbb{P} \left( \text{ exists } t \in \mathbb{N}^+ : \theta_* \notin \mathcal{C}_t \right) \leq \delta$ with*

$$\mathcal{C}_t = \left\{ \theta \in \mathbb{R}^d : \left\| \hat{\theta}_{t-1} - \theta \right\|_{V_{t-1}(\lambda)} < L\sqrt{\lambda} + \sqrt{2 \log \left( \frac{1}{\delta} \right) + \log \left( \frac{\det V_{t-1}(\lambda)}{\lambda^d} \right)} \right\}.$$

## 6.2 Proofs of Main Theorems

In this section, we prove that the algorithm in 2 satisfies the regret bound given in Theorem 3 Consider the 'good' event $E$ defined as follows

$$E = \bigcap_{t=1}^{T} \bigcap_{i=1}^{m} \left\{ \boldsymbol{\theta}_t^i \in C_t(\boldsymbol{\theta^*}) \right\} \tag{15}$$

This event states that at every round $t \in [T]$, every player $i \in [m]$ has an empirical estimate of $\boldsymbol{\theta^*}$ that is within the confidence interval centered at $\boldsymbol{\theta^*}$. This ensures that all of the player's estimates of $\boldsymbol{\theta^*}$ are not too far from each other. This also means that despite each player having a different empirical estimate of $\boldsymbol{\theta^*}$, if the context vectors of each arm are not too close for most rounds, then the players will be able to coordinate properly. This is formalized in lemma 7. To do that we first show that the eigenvalues of $V_t$ are nondecreasing

**Lemma 6.** *For any $\lambda > 0$ and $\beta_T$, we have the following inequality for each players estimate for $\boldsymbol{\theta}_t^i$ and $\theta^*$*

$$\left\| \boldsymbol{\theta}_t^i - \boldsymbol{\theta^*} \right\| \leq \frac{\beta_T}{\lambda}$$

*Proof.* To prove this note that $C_t$ is an ellipsoid where the inverse of the eigenvalues of $V_{t-1}$ give the lengths of the principle axes. We first note that based on the fact that $V_0 = \lambda I$, and therefore $C_0$ is a circle with radius $\frac{\beta_T}{\lambda}$. We will be done if we can show that $V_t$ has *nondecreasing* eigenvalues. Let $\sigma_1^k \geq \sigma_2^k \geq \cdots \geq \sigma_t^d$ be the eigenvalues of $V_t$.

From the definition of $V_t$, it's clear that $V_t$ is symmetric. Thus, we can apply the Courant-Fischer min-max Theorem to obtain

$$\sigma_t^k(A) = \min\{\max\{R_{V_t}(\boldsymbol{v}) \mid \boldsymbol{v} \in U \text{ and } \boldsymbol{v} \neq 0\} \mid \dim(U) = k\}$$

where the Rayleigh Quotient $R_{V_t}(\boldsymbol{v})$ is defined as,

$$R_{V_t}(\boldsymbol{v}) = \frac{\langle V_t \boldsymbol{v}, \boldsymbol{v} \rangle}{\left\| \boldsymbol{v} \right\|^2}$$

Therefore, we have

$$
\begin{aligned}
\sigma_{t+1}^k &= \min\{\max\{R_{V_{t+1}}(x) \mid \boldsymbol{v} \in U, v \neq 0\} \mid \dim(U) = k\} \\
&= \min\{\max\{R_{V_t + \boldsymbol{x}_{t,\boldsymbol{a}_t}\boldsymbol{x}_{t,\boldsymbol{a}_t}^\top}(x) \mid \boldsymbol{v} \in U, v \neq 0\} \mid \dim(U) = k\} \\
&= \min\left\{ \max\left\{ \frac{\langle(V_t + \boldsymbol{x}_{t,\boldsymbol{a}_t}\boldsymbol{x}_{t,\boldsymbol{a}_t}^\top)\boldsymbol{v}, \boldsymbol{v}\rangle}{\|\boldsymbol{v}\|^2} \middle| \boldsymbol{v} \in U, v \neq 0 \right\} \right. \\
&\qquad \left. \middle| \dim(U) = k \right\} \\
&= \min\left\{ \max\left\{ \frac{\langle V_t\boldsymbol{v}, \boldsymbol{v}\rangle}{\|\boldsymbol{v}\|^2} + \frac{\langle \boldsymbol{x}_{t,\boldsymbol{a}_t}\boldsymbol{x}_{t,\boldsymbol{a}_t}^\top\boldsymbol{v}, \boldsymbol{v}\rangle}{\|\boldsymbol{v}\|^2} \middle| \boldsymbol{v} \in U, v \neq 0 \right\} \right. \\
&\qquad \left. \middle| \dim(U) = k \right\} \\
&\geq \min\left\{ \max\left\{ \frac{\langle V_t\boldsymbol{v}, \boldsymbol{v}\rangle}{\|\boldsymbol{v}\|^2} \middle| \boldsymbol{v} \in U, v \neq 0 \right\} \middle| \dim(U) = k \right\} \\
&= \sigma_t^k
\end{aligned}
$$

where in the inequality we used the fact that $\langle \boldsymbol{x}_{t,\boldsymbol{a}_t}\boldsymbol{x}_{t,\boldsymbol{a}_t}^\top\boldsymbol{v}, \boldsymbol{v}\rangle = (\boldsymbol{x}_{t,\boldsymbol{a}_t}\boldsymbol{x}_{t,\boldsymbol{a}_t}^\top v)^\top v = v^\top \boldsymbol{x}_{t,\boldsymbol{a}_t}\boldsymbol{x}_{t,\boldsymbol{a}_t}^\top v = \|\boldsymbol{x}_{t,\boldsymbol{a}_t}^\top v\| \geq 0$. $\qquad\square$

Now we show that when all the players have their estimates inside the confidence ball around $\boldsymbol{\theta}^*$, then they can fully coordinate.

**Lemma 7.** *Suppose $\boldsymbol{\theta}_t^i \in C_t(\theta)$ is an empirical estimate of $\boldsymbol{\theta}^*$ for players $i$. Then under the good event $E$, if $\boldsymbol{x}_{t,\boldsymbol{a}}$ and $\boldsymbol{x}_{t',\boldsymbol{a}'}$ are context vectors such that*

$$
\langle \boldsymbol{\theta}^*, \boldsymbol{x}_{t,\boldsymbol{a}}\rangle - \langle \boldsymbol{\theta}^*, \boldsymbol{x}_{t,\boldsymbol{a}'}\rangle > 2\frac{\beta_T L}{\lambda} \tag{16}
$$

*then $\langle \boldsymbol{\theta}_t^i, \boldsymbol{x}_{t,\boldsymbol{a}}\rangle > \langle \boldsymbol{\theta}_t^i, \boldsymbol{x}_{t,\boldsymbol{a}'}\rangle$ for all players $i$.*

*Proof.* From the defintion of $C_t(\theta)$, we know that

$$
\begin{aligned}
\langle \boldsymbol{\theta}_t^i, \boldsymbol{x}_{t,\boldsymbol{a}'}\rangle &= \langle \boldsymbol{\theta}^*, \boldsymbol{x}_{t,\boldsymbol{a}'}\rangle + \langle \boldsymbol{\theta}_t^i - \boldsymbol{\theta}^*, \boldsymbol{x}_{t,\boldsymbol{a}'}\rangle &&(17) \\
&\leq \langle \boldsymbol{\theta}^*, \boldsymbol{x}_{t,\boldsymbol{a}'}\rangle + \|\boldsymbol{\theta}_t^i - \boldsymbol{\theta}^*\|\|\boldsymbol{x}_{t,\boldsymbol{a}'}\| &&(18) \\
&\leq \langle \boldsymbol{\theta}^*, \boldsymbol{x}_{t,\boldsymbol{a}'}\rangle + \|\boldsymbol{\theta}_t^i - \boldsymbol{\theta}^*\|L &&(19)
\end{aligned}
$$

Similarly,

$$
\begin{aligned}
\langle \boldsymbol{\theta}_t^i, \boldsymbol{x}_{t,\boldsymbol{a}}\rangle &= \langle \boldsymbol{\theta}^*, \boldsymbol{x}_{t,\boldsymbol{a}}\rangle + \langle \boldsymbol{\theta}_t^i - \boldsymbol{\theta}^*, \boldsymbol{x}_{t,\boldsymbol{a}}\rangle &&(20) \\
&\geq \langle \boldsymbol{\theta}^*, \boldsymbol{x}_{t,\boldsymbol{a}}\rangle - \|\boldsymbol{\theta}_t^i - \boldsymbol{\theta}^*\|\|\boldsymbol{x}_{t,\boldsymbol{a}}\| &&(21) \\
&\geq \langle \boldsymbol{\theta}^*, \boldsymbol{x}_{t,\boldsymbol{a}}\rangle - \|\boldsymbol{\theta}_t^i - \boldsymbol{\theta}^*\|L &&(22)
\end{aligned}
$$

Therefore combining the two inequalities above yields

$$
\begin{aligned}
\langle \boldsymbol{\theta}_t^i, \boldsymbol{x}_{t,\boldsymbol{a}}\rangle - \langle \boldsymbol{\theta}_t^i, \boldsymbol{x}_{t,\boldsymbol{a}'}\rangle &\geq \langle \boldsymbol{\theta}^*, \boldsymbol{x}_{t,\boldsymbol{a}}\rangle - \|\boldsymbol{\theta}_t^i - \boldsymbol{\theta}^*\|L \\
&\quad - \left(\langle \boldsymbol{\theta}^*, \boldsymbol{x}_{t,\boldsymbol{a}'}\rangle + \|\boldsymbol{\theta}_t^i - \boldsymbol{\theta}^*\|L\right) \\
&\geq \langle \boldsymbol{\theta}^*, \boldsymbol{x}_{t,\boldsymbol{a}}\rangle - \langle \boldsymbol{\theta}^*, \boldsymbol{x}_{t,\boldsymbol{a}'}\rangle \\
&\quad - 2\|\boldsymbol{\theta}_t^i - \boldsymbol{\theta}^*\|L
\end{aligned}
$$

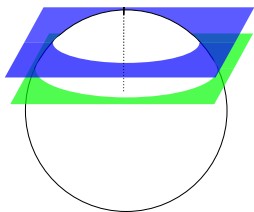

Figure 2: The set of points $\boldsymbol{x}_{\boldsymbol{a}_2}$ such that equation 23 is satisfied lies outside of the region bounded by the blue and green hyperplanes determined by $\boldsymbol{x}_{t,\boldsymbol{a}_t}$. The dotted vector is $\boldsymbol{\theta}^*$, and these hyperplanes are a distance of $4\frac{\beta_T L}{\lambda \|\boldsymbol{\theta}^*\|}$ apart.

Form Lemma 6, $\left\|\boldsymbol{\theta}_t^i - \boldsymbol{\theta}^*\right\| \leq \frac{\beta_T}{\lambda}$, then equation equation 16 will show $\langle \boldsymbol{\theta}^*, \boldsymbol{x}_{t,\boldsymbol{a}} \rangle - \langle \boldsymbol{\theta}^*, \boldsymbol{x}_{t,\boldsymbol{a}'} \rangle - 2\left\|\boldsymbol{\theta}_t^i - \boldsymbol{\theta}^*\right\| L > 0$ and the desired result will follows.

This proves the desired result. $\qquad\square$

The next result tells us that the probability that the context vectors satisfy the hypothesis in Lemma 7 is lower bounded by some constant that will grow to 1 as $T \to \infty$. This will be used to define the good event $G_t$ that will allow the players to agree on which arm they want to pull.

**Lemma 8.** *At any given round $t$, if all the context vectors $\boldsymbol{x}_{t,\boldsymbol{a}}$ are generated at random with probability density function $\psi(x) < M$, with $\|\boldsymbol{x}_{t,\boldsymbol{a}}\| \leq L$, Then let $P_t$ be the probability for the following event at a round $t$: Any two joint actions $\boldsymbol{a}$ and $\boldsymbol{a}'$ satisfies the following inequality*

$$|\langle \boldsymbol{\theta}^*, \boldsymbol{x}_{t,\boldsymbol{a}} \rangle - \langle \boldsymbol{\theta}^*, \boldsymbol{x}_{t,\boldsymbol{a}'} \rangle| > 2\frac{\beta_T L}{\lambda} \tag{23}$$

*Then*

$$P_t \geq 1 - K^{2m}\frac{c_2 M (c_1 L)^d \beta_T}{\lambda} \tag{24}$$

*for universal constants $c_1, c_2 \in \mathbb{R}$.*

*Proof.* Aribtrailiry order the joint actions as $\boldsymbol{a}_1, \boldsymbol{a}_2, ...$, and suppose $\boldsymbol{x}_{t,\boldsymbol{a}_1}$ has been placed so that the given conditions are satisfied. Now let's bound the volume where the next context vector can be placed. In particular, the set of points $\boldsymbol{x}_{t,\boldsymbol{a}_2}$ such that it satisfies equation 23 satisfies

$$\langle \boldsymbol{\theta}^*, \boldsymbol{x}_{t,\boldsymbol{a}_2} \rangle > \langle \boldsymbol{\theta}^*, \boldsymbol{x}_{t,\boldsymbol{a}_1} \rangle + 2\frac{\beta_T L}{\lambda} \quad \text{or} \tag{25}$$

$$\langle \boldsymbol{\theta}^*, \boldsymbol{x}_{t,\boldsymbol{a}_2} \rangle < \langle \boldsymbol{\theta}^*, \boldsymbol{x}_{t,\boldsymbol{a}_1} \rangle - 2\frac{\beta_T L}{\lambda} \tag{26}$$

From the definition of the inner product, the set of $\boldsymbol{x}_{t,\boldsymbol{a}_2}$ that satisfy the equation above lies outside of two hyperplanes normal to $\boldsymbol{\theta}^*$ and at a distance of $4\frac{\beta_T L}{\lambda \|\boldsymbol{\theta}^*\|}$ apart. See Figure 2 for an example in $d = 3$. Call the region between these two parallel hyperplanes contained within the sphere $U$. Then the volume of $U$ can be bounded by the volume of a cylinder whose base is an $d - 1$ dimensional sphere with radius $L$, and with height $4\frac{\beta_T L}{\lambda \|\boldsymbol{\theta}^*\|}$. Thus the volume of each such region is upper bounded by

$$\mu(U) = \frac{\pi^{\frac{d-1}{2}}}{\Gamma(\frac{d+1}{2})} L^{d-1}\left(4\frac{\beta_T L}{\lambda \|\boldsymbol{\theta}^*\|}\right) = \frac{\pi^{\frac{d-1}{2}}}{\Gamma(\frac{d+1}{2})} L^d\left(4\frac{\beta_T}{\lambda \|\boldsymbol{\theta}^*\|}\right)$$

Thus the probability that $\boldsymbol{x}_{t,\boldsymbol{a}_2}$ satisfies equation 23 is at least

$$1 - \int_U \psi(x)dx \geq 1 - \mu(U)M \tag{27}$$

$$\geq 1 - M\pi^{\frac{d-1}{2}}L^d\left(4\frac{\beta_T}{\lambda\|\boldsymbol{\theta^*}\|}\right) \tag{28}$$

$$\geq 1 - \frac{c_2 M(c_1 L)^d \beta_T}{\lambda} \tag{29}$$

for some universal constants $c_1, c_2$. Repeating inductively, the probability that all $K^m$ context vectors satisfy equation 23 is at least

$$\prod_{k=1}^{K^m}\left(1 - k\frac{c_2 M(c_1 L)^d \beta_T}{\lambda}\right) \geq \left(1 - K^m \frac{c_2 M(c_1 L)^d \beta_T}{\lambda}\right)^{K^m} \tag{30}$$

$$\geq 1 - K^{2m}\frac{c_2 M(c_1 L)^d \beta_T}{\lambda} \tag{31}$$

where in the last inequality we used $(1-x)^n \geq 1 - nx$ for $x \geq 0$. $\qquad\square$

**Theorem 4** In the reward and action asymmetric (problem C) contextual bandit setting where the context vectors are distributed with fixed distribution the frequentist regret bound of the algorithm is

$$R_T = O(mK^{2m}L^d\sqrt{T}\log(T)) \tag{32}$$

*Proof.* Consider the 'good' event at time $t$ defined as,

$$G_t = \bigcap_{\boldsymbol{a},\boldsymbol{a}'\in\mathcal{A}}\left\{|\langle\boldsymbol{\theta^*},\boldsymbol{x}_{t,\boldsymbol{a}}\rangle - \langle\boldsymbol{\theta^*},\boldsymbol{x}_{t,\boldsymbol{a}'}\rangle| > 2\frac{\beta_T}{L}\right\}$$

and let

$$G = \bigcap_{t=1}^{T}G_t$$

This is the event that at round $T$, the context vectors for any two joint actions $\boldsymbol{a}$ and $\boldsymbol{a}'$ are not too close in the sense that their inner product with $\boldsymbol{\theta^*}$ is sufficiently far.

We suppose there are $T^\alpha$ rounds of exploration for some $\alpha \in (0,1)$ and then optimize over $\alpha$. We can decompose the regret as follows:

$$R_T = \mathbb{E}\left[\sum_{t=1}^{T} \langle \boldsymbol{\theta}, \boldsymbol{x}_{\boldsymbol{a}_t} - \boldsymbol{x}^* \rangle\right] \tag{33}$$

$$\leq \mathbb{E}\left[\sum_{t=1}^{T^\alpha} \langle \boldsymbol{\theta}, \boldsymbol{x}_{\boldsymbol{a}_t} - \boldsymbol{x}^* \rangle + \sum_{t=T^\alpha}^{T} \langle \boldsymbol{\theta}, \boldsymbol{x}_{\boldsymbol{a}_t} - \boldsymbol{x}^* \rangle\right] \tag{34}$$

$$\leq O(T^\alpha) + \mathbb{E}\left[\sum_{t=T^\alpha}^{T} \langle \boldsymbol{\theta}, \boldsymbol{x}_{\boldsymbol{a}_t} - \boldsymbol{x}^* \rangle\right] \tag{35}$$

$$= O(T^\alpha) + \mathbb{E}\left[\sum_{t=T^\alpha}^{T} \langle \boldsymbol{\theta}, \boldsymbol{x}_{\boldsymbol{a}_t} - \boldsymbol{x}^* \rangle (\mathbb{I}[G_t \cap E] + \mathbb{I}[(G_t \cap E)^c]\right] \tag{36}$$

$$= O(T^\alpha) + \mathbb{E}\left[\sum_{t=T^\alpha}^{T} \langle \boldsymbol{\theta}, \boldsymbol{x}_{\boldsymbol{a}_t} - \boldsymbol{x}^* \rangle (\mathbb{I}[G_t \cap E] + \mathbb{I}[(G_t \cap E)^c]\right] \tag{37}$$

$$= O(T^\alpha) + \mathbb{E}\left[\sum_{t=T^\alpha}^{T} \langle \boldsymbol{\theta}, \boldsymbol{x}_{\boldsymbol{a}_t} - \boldsymbol{x}^* \rangle \mathbb{I}[G_t \cap E]\right] + \sum_{t=T^\alpha}^{T} P(G_t \cap E)^c \tag{38}$$

$$\leq O(T^\alpha) + \mathbb{E}\left[\sum_{t=T^\alpha}^{T} \langle \boldsymbol{\theta}, \boldsymbol{x}_{\boldsymbol{a}_t} - \boldsymbol{x}^* \rangle \mathbb{I}[G_t \cap E]\right] + \sum_{t=T^\alpha}^{T} [P(G_t^c) + P(E^c)] \tag{39}$$

After $T^\alpha$ rounds of exploration, we have $\lambda = T^\alpha$. Furthermore, as in Lemma 5, the probability that for all players $i, \in [M]$ their estimator is within the confidence interval (determined by $\beta_T$) is at least $1 - \delta$. Thus the probability that everyone's estimator is within this confidence interval is $1 - m\delta$. Picking $\delta = \frac{1}{T}$ this gives

$$P(E^c) \leq m\delta = \frac{m}{T}$$

Using our choices of $\delta$ and $\lambda = T^\alpha$, we have (by Theorem 19.2 of Lattimore and Szepesvári (2020))

$$\sqrt{\beta_T} = \sqrt{\lambda}L + \sqrt{2\log(T) + \log\left(\frac{\det(V_T(\lambda))}{\lambda^d}\right)} = T^{\alpha/2}L + \sqrt{2\log(T) + \log\left(\frac{d\lambda + TL^2}{T^{\alpha d}}\right)}$$

So that we can use lemma 8 to upper bound the probability of the complement of the good event happening, we have

$$P(G_t^c) \leq K^{2m}\frac{c_2 M(c_1 L)^d \beta_T}{\lambda} = K^{2m}\frac{c_2 M(c_1 L)^d \left[2\log(T) + \log\left(\frac{\det(V_T(\lambda))}{T^{\alpha d}}\right)\right]}{T^\alpha} = O\left(K^{2m}\frac{L^d \log(T)}{T^\alpha}\right) \tag{40}$$

According to Lemma 7, under the event $\mathbb{I}[G_t \cap E]$ the players are completely coordinated. This means that we are reduced to a single agent setting with a $K^m$ size action space. However, the bound for the single agent LinUCB regret bound doesn't depend on the size of the action space so we do not expect the exponentially larger action space to affect the regret. The regret for this can be bounded as follows.

Let $r_t$ be the instantaneous regret in round $t$ (under the good event $G_t \cap E$ defined by,

$$r_t = \langle \theta_*, \boldsymbol{x}_{\boldsymbol{a}_t^*} - \boldsymbol{x}_{\boldsymbol{a}_t} \rangle.$$

where $\boldsymbol{a}_t^*$ is the optimal arm for round $t$ based on the context vectors received. Let $\tilde{\theta}_t \in \mathcal{C}_t$ be the parameter in the confidence set for which $\langle \tilde{\theta}_t, \boldsymbol{a}_t \rangle = \text{UCB}_t(\boldsymbol{a}_t)$. Then, using the fact that $\theta_* \in \mathcal{C}_t$ and the definition of the algorithm leads to

$$\langle \theta_*, \boldsymbol{x}_{\boldsymbol{a}_t^*} \rangle \leq \text{UCB}_t(\boldsymbol{x}_{\boldsymbol{a}_t^*}) \leq \text{UCB}_t(\boldsymbol{x}_{\boldsymbol{a}_t}) = \langle \tilde{\theta}_t, \boldsymbol{x}_{\boldsymbol{a}_t} \rangle.$$

Using Cauchy-Schwarz inequality and the assumption that $\theta_* \in \mathcal{C}_t$ and facts that $\tilde{\theta}_t \in \mathcal{C}_t$ and $\mathcal{C}_t \subseteq \mathcal{E}_t$ leads to

$$r_t = \left\langle \theta_*, \boldsymbol{x}_{\boldsymbol{a}_t^*} - \boldsymbol{x}_{\boldsymbol{a}_t} \right\rangle \leq \left\langle \tilde{\theta}_t - \theta_*, \boldsymbol{x}_{\boldsymbol{a}_t} \right\rangle \leq \left\| \boldsymbol{x}_{\boldsymbol{a}_t} \right\|_{V_T^{-1}} \left\| \tilde{\theta}_t - \theta_* \right\|_{V_T} \tag{41}$$

$$\leq 2 \left\| \boldsymbol{x}_{\boldsymbol{a}_t} \right\|_{V_T^{-1}} \sqrt{\beta_T} = \boldsymbol{x}_{\boldsymbol{a}_t}^\top V_T^{-1} \boldsymbol{x}_{\boldsymbol{a}_t} \sqrt{\beta_T} \leq O\left( 2 \frac{L^2 \sqrt{\beta_T}}{T^\alpha} \right) \tag{42}$$

Where use used the fact that $\left\| V_T^{-1} \right\| = \max_{x \in \mathbb{R}^d} \frac{\left\| V_T^{-1} x \right\|_2}{\left\| x \right\|_2}$ is upper bounded by the largest eigenvalue $= O(\frac{1}{T^\alpha})$ (given by $\frac{1}{\lambda}$) since $V_t^{-1}$ is positive semidefinite.

Therefore, picking $\alpha = \frac{1}{2}$ which gives us the tightest bound by AM-GM, we have

$$R_T = \sum_{t=1}^{T} [P(G_t^c) + P(E^c)] + \sum_{t=1}^{T} r_t = O(mK^{2m} L^d \sqrt{T} \log(T)) \tag{43}$$

$\qquad\qquad\qquad\qquad\qquad\qquad\qquad\qquad\qquad\qquad\qquad\qquad\qquad\qquad\qquad\qquad\qquad\square$

We can now prove the regret bound of Algorithm 2 under reward asymmetry (Problem B).

**Theorem 3** In the reward asymmetric (Problem B) contextual bandit setting where the context vectors are distributed with fixed distribution the frequentist regret bound of Algorithm 2 is

$$R_T = O(mK^{2m} L^d \sqrt{T} \log(T)) \tag{44}$$

*Proof.* Remarkably we can follow the same proof structure as in Theorem 4. In `ETC` we have two main phases

1. They pull a fixed arbitrary arm for $T^\alpha$ exploration rounds while updating their $\hat{\theta}$ estimate.

2. The remaining $T - T^\alpha$ rounds they will do regular Lin-UCB while not updating their parameters.

Even though in Algorithm `LinUCB-B`, they follow LinUCB for all $T$ rounds, however, we can decompose these rounds into the set of first $T^\alpha$ rounds and the remaining $T - T^\alpha$ rounds to capitalize on the decomposition given by equation 39. This is because in the first $T^\alpha$ rounds we are still updating our estimate for $\hat{\theta}$ which is exactly what happens in phase 1 of `ETC`. Given that our initialization $\lambda = T^\alpha$ is unchanged in `LinUCB-B` from `ETC` this means that equation 40 still holds. While for the remaining $T - \hat{\theta}$ rounds they will do regular Lin-UCB while still sharpening the parameters which is essentially a better version of phase 2 of `ETC`. This means that equation 42 still holds. In fact, this equation can be made slightly sharper by

$$r_t \leq O\left( 2 \frac{L^2 \sqrt{\beta_T}}{t} \right) \tag{45}$$

Therefore we will obta ain sharper bound but of the same order as $O(\cdot)$ hides the constants. $\qquad\square$

