# OpenReview forum: "Multiplayer Information Asymmetric Contextual Bandits"
_TMLR — Accepted by TMLR_

### Review · Reviewer_Ytk6 · 2024-11-26

**Summary Of Contributions:**

This paper introduces a formulation and associated algorithms for multiplayer contextual bandits under action and/or reward information asymmetries. The authors propose adaptations of the LinUCB algorithm tailored to this setting, present regret bounds and experimental results to support the theoretical arguments. For the case of action asymmetry, they demonstrate that the regret achieves the optimal sublinear bound of $O(\sqrt{T})$ Additionally, for reward asymmetry, and action and reward asymmetry, they present the first sublinear regret bounds, with polynomial dependency on the size of the joint action space.

**Audience:**

Yes

**Claims And Evidence:**

Yes

**Requested Changes:**

Improve the experiments section as in above
Proofread the paper

**Strengths And Weaknesses:**

Strengths
In addition to being the first work to study multiplayer contextual bandits, the authors have investigated information asymmetry under action and/or reward information asymmetries, and provided a well-thought-out presentation under the 3 cases. The disposition is clear and coherent, and the theoretical results and algorithms have been presented concisely and discussed well.

Weaknesses
- The regret bounds for Problems B and C scale poorly with the actions $K$, and players $m$, potentially making practical applications challenging. For instance, even with moderate values like $K = 10$ and $m = 4$, the polynomial dependency leads to high regret. It would be beneficial to investigate the empirical consistency of these bounds to assess their practical relevance.
-  The purpose and goals of the experiments are not clearly stated. Providing a detailed introduction to the experimental setup, including the specific objectives and hypotheses being tested, would improve clarity.
- The experiments primarily focus on small configurations of $d$, $K$, and $m$. Expanding the investigation to include higher-dimensional contexts ($d$), larger action spaces ($K$), and more players ($m$) would help validate the theoretical bounds and offer deeper insights into the scalability and robustness of the algorithms.
- It is unclear which baselines, if any, are being compared. Explicitly include comparisons with either naively constructed baselines or existing algorithms for multiplayer bandits, to contextualize the performance of the proposed methods.
- There also seems to be quite a lot of typos and hanging sentences in the paper. Please do a comprehensive proofread of the paper, particularly in the Analysis and conclusion.

Other comments: Have you considered selecting $\lambda$ in the ETC algorithm dynamically e.g. stopping exploration when confidence intervals shrink sufficiently?

---

> ### Author Response · Authors · 2024-12-05
> **Response to Reviewer Ytk6**
>
> Thanks for taking the time to read our paper and for the valuable feedback. Our responses are below
> - Indeed, because of the necessity of coordination, our algorithm scales poorly with K and M. It may be possible with some stronger assumptions on the context vectors that we can obtain a bound that doesn't scale poorly with these 2 parameters. But for this paper, we wanted to provide a sublinear regret bound (in T) for the first information asymmetric multiplayer contextual bandits problem in a more general setting. It would be interesting as future work to see how we can improve on the dependence of K and M.
>
> - The purpose of the experiments was to compare the asymptotic behavior of the regret of our algorithms (in T) with the theoretical bounds provided. We apologize for not making this more clear; we have modified this in our manuscript.
>
> - Due to computation limitations we only looked at small values of K, m, and d. We understand that there is a line of work that studies regret bounds with larger values of d, and this would definitely be interesting to study for future work.
>
> - The closest thing to a "baseline" would be our algorithm for Problem A since it is nearly the same as the single agent setting but with an ordering on the joint actions agreed upon by the players prior to the start of learning. We do include that in Figure 1 (it's in color red). However, there aren't any other algorithms for this setting since the multiplayer information asymmetric setting wasn't studied until recently. We changed it to make it more clear that the plot for Problem A is our "baseline" in the final version of the paper.
>
> - We apologize for the typos in the analysis and conclusion paper. We will thoroughly proofread the final version.
>
> - Regarding the "other comment" due to the fact that in problem C the players cannot observe actions AND get i.i.d. copies of the reward, trying to dynamically stop exploration would be tricky since different players may stop at different times. This may be less of an issue in problem B where the actions can be observed, although currently, the efficacy is still unclear. We appreciate this suggestion and will include a discussion of this in the final version and perhaps delegate it for appropriate future work.

---

### Review · Reviewer_4zmn · 2024-12-01

**Summary Of Contributions:**

This paper studied the contextual bandits where the multiple players obtain asymmetric information. It considered there different settings: (a) an agent can observe others' actions but only their own i.i.d. reward; (b) an agent cannot observe others' actions but their unique reward; (c) an agent cannot observe others' actions and their own i.i.d. reward. It derived the regret of proposed algorithms and evaluated them with simulations.

**Audience:**

No

**Broader Impact Concerns:**

No ethical concern.

**Claims And Evidence:**

No

**Requested Changes:**

Suggestions are in the section of 'Strengths And Weaknesses'.

**Strengths And Weaknesses:**

Strengths:
1. The LinUCB algorithm is described in Section 3.1 in detail.
1. Some related works are discussed.

Weaknesses:
1. What is the motivation of the several multiplayer settings?
1. Are there lower bounds to show the efficiency of proposed algorithms?
1. Consider the similarity to Chang et al. (2022), can the algorithms be compared in some setting?
1. In Section 2, it is stated that 'the case where each player has a different number of arms is easy'. Why?
1. I suggest the author(s) to provide the proof of Theorem 2.
1. The design of Algorithms 1 and 2 look confusing. It is not easy to understand the pseudocodes.
1. It is better to clearly describe the formulation of problem B.
1. The setting studied in section 3.4 can be mentioned in Section 1.

Other comments: The author(s) can more careful about the writing. Some suggestions are as below.
1. can be improved: 'Furthermore, at the start of each round, every player is given the same $K^m$ context vectors $x_{t,a} \in R^d$ corresponding to each joint arm $a \in A$.
1. can be improved: 'Each player maintains for each arm their own UCB-interval and when two UCB-intervals are disjoint.'
1. can be improved: 'This algorithm is similar as mLinUCB-A but takes into account that the reward feedback is different for different players'?
1. Title of Section 3.2: 'Asymmetry in actions'?

---

> ### Author Response · Authors · 2024-12-05
> **Response to Reviewer 4zmn**
>
> Thanks for taking the time to read our paper and for the valuable feedback. Our responses are below
>
> 1. The multiplayer information asymmetric bandit setting is relatively new and interesting and there have been works on the theory of this problem (e.g. Chang 2022). Since the single agent contextual bandit problem has been extensively and rife with applications (e.g. recommender systems). Please see Bouneffouf, Djallel, and Irina Rish. "A survey on practical applications of multi-armed and contextual bandits." (2019) for more details. As a result, we believe our paper naturally extends the applications previously studied already to allow for multiple learners allowing for information asymmetry.
>
> 2. The lower bounds are similar to the single-agent setting since the reward structure is the same (the main difference is the information asymmetry between players). We will mention the lower bounds from the single-player setting and compare them with our theorems in the final version.
>
> 3. The algorithms in chang 2022 are for standard MAB problems which are difficult to compare to the contextual bandit problem. The closest thing to a "baseline" would be our algorithm for Problem A since it is nearly the same as the single agent setting but with an ordering on the joint actions agreed upon by the players prior to the start of learning. We do include that in Figure 1 (it's in color red). However, there aren't any other algorithms for this setting since the multiplayer information asymmetric setting wasn't studied until recently. We have modified our manuscript to make it clear that our plot for Problem A is our "baseline" in the final version of the paper.
>
> 4. Apologies for being unclear; we meant to say that to extend our algorithms to the case where the number of arms is different is simple. As long as the players know how many actions the other players have prior to the learning, they are able to apply the same coordination principles in this paper to achieve the same order regret bound in T. We will definitely fix this in the final version
>
> 5. Since we only added a coordination scheme, the proof of the regret bound is the same as the single agent setting given in Wei Chu et al. “Contextual bandits with linear payoff functions” 2014. The main difference is just changing the notation. We will include more details (at least the key ingredients of the proof) in the final version.
>
> 6. We apologize for the confusion, if there is a specific part that is confusing we would be happy to modify it for more clarity. If it helps, the pseudocode structure is similar to Wei Chu et al. “Contextual bandits with linear payoff functions” 2014.
>
> 7. We understand the setting is relatively new and there are few prior works. We have added additional details to the statement of problem B.
>
> 8. This section is for Problem C, which is discussed in the preliminary. We have modified the manuscript to make this more clear.
>
> Other comments:
> Thank you for pointing out our errors, we have modified them in our manuscript and hope that these changes will improve the paper. In particular, we have modified the second bullet point to be "In that paper, each player maintains for each arm their own UCB-interval, and when two UCB intervals are disjoint the suboptimal arm gets eliminated." The other sentences have been modified accordingly.

---

### Review · Reviewer_KpK6 · 2024-12-11

**Summary Of Contributions:**

This paper formulates a new multi-agent contextual bandit problem. It considers three types of asymmetry scenarios, the action, the reward, and both. For each asymmetry scenario, the paper provides a variant of LinUCB algorithm. For the action scenario, it proves an $O(\sqrt{T})$ regret. For the latter two scenarios, it proves an $O(mK^{2m}L^d\sqrt{T})$ regret.

**Audience:**

Yes

**Claims And Evidence:**

Yes

**Requested Changes:**

Please see the last part.

**Strengths And Weaknesses:**

The following are some questions:
1. There are a rich line of works studying the multi-agent bandit problem. In some cases, the agents are cooperative and aim to maximize the collective rewards. In some cases, the agents are competitive and aim to maximize their own rewards. And also some are homogeneous setting where agents' preferences are the same and some are heterogeneous where agents' preferences are different. This paper does not discuss these related works. And in these works, usually single agent can also only observe their own rewards. What is the difference between the considerred setting with existing setting? Is the proposed framework more general and can cover existing setting?
2. For the action asymmetry setting, single agent can only observe its own action but cannot observe other agents' actions. In such a case, why the agent can observe the full $x_{t,a}$ context where $a$ is the joint action over all agents.
3. The theoretical significance is limited. The analysis in the action asymmetry setting is a direct extension of previous LinUCB algorithm only a minor modification of how to break the tie. For the reward asymmetry setting, the regret has an exponential term which does not appear in existing multi-agent bandit works.
4. Some typos: the second paragraph in page 4 "each player is observes", "Each player maintains for each arm their own UCB-interval and when two UCB-intervals are disjoint. However, such an elimination method..."; The last paragraph in page 10 "a $O(T \sqrt{T})$ regret bound", "the first $\sqrt{T}$ rounds".
5. I am not convinced that "However, this problem is easier than the bandits’ case in the sense that we only have to estimate one parameter θ ∗ , whereas in the bandits case, we had to estimate the mean of each arm." Usually, due to limited observation, the linear bandit problem is much harder than the tabular multi-armed bandit setting. Many problems are first solved in the tabular setting and then the fuction approximation setting (such as the linear bandit setting) are solved.

---

> ### Author Response · Authors · 2024-12-12
> **Response to Reviewer KpK6**
>
> Dear reviewer KpK6,
>
> Thank you very much for bringing up these insightful comments. Here are our responses:
>
> 1. We apologize for not including a more thorough review of the multiplayer MAB works. To the best of our knowledge, there are no existing papers for the multiplayer contextual bandit's settings. As far as the standard MAB is concerned, while there are a lot of works for the multiplayer settings, they usually do not consider joint actions or they allow communication. For example, in this paper on cooperative bandits (Dubey 2020, https://proceedings.mlr.press/v119/dubey20a/dubey20a.pdf) the player plays on the same set of actions whereas we consider joint actions. Another example would be the heterogeneous setting (Xu, 2023 https://proceedings.neurips.cc/paper_files/paper/2023/file/ec795aeadae0b7d230fa35cbaf04c041-Paper-Conference.pdf) where they have a graph structure that allows them to communicate with other players. In this regard, our study proposes challenges that their papers do not have. This is why we believe these multiagent settings are not comparable to ours. In the final version, we will include a more thorough review of these settings.
>
> 2. We were trying to model multiplayer information asymmetric contextual bandits as closely as we can to the previous theoretical works on contextual bandits, where it is necessary for the learner to observe a context vector for every arm. Without the ability to observe context vectors, contextual bandits would not be a solvable problem. Having said that, we agree that it is interesting to study a generalization of our setting where each player observes a noisy version of the context vectors.
>
> 3. We wanted to show that there is a very natural extension of the LinUCB algorithm to the multiplayer contextual setting. We believe that this natural and yet simple extension is an asset for the paper rather than a drawback. That is, we can apply existing works that are theoretically rich to a novel setting. For the information asymmetric setting, while it is true that there is an exponential term which is not ideal, this is still the first sub-linear regret bound (in time) that exists for this setting. The difficulty in removing the exponential term is that it exists because of the necessity of coordination with other players in this setting (not to mention the information asymmetry), which does not exist in the single-player setting. We believe that trying to remove this exponential factor would be interesting for future works.
>
> 4. Thank you very much for pointing this out to us. We will fix this in the final version of the paper.
>
> 5. Apologies for not making this more clear. We were trying to say that the multi-agent extension of the standard MAB (Chang 2023, https://arxiv.org/abs/2311.06210) is more difficult than the multi-agent extension for contextual bandits from an algorithmic standpoint. The algorithm that we use in our paper for the multiplayer contextual bandits setting is only small modifications from the LinUCB algorithm of the single-player setting, which we again consider as an asset for our paper instead of a drawback. In the standard MAB, the challenge of extending it to a multiplayer setting is that miscoordination often leads to linear regret. However, in the contextual bandits setting, since there is only one parameter to estimate (\theta) so miscoordination does not always lead to linear regret, which is one of the contributions of our paper as well.

---

### Decision · Action_Editor_ymZw · 2025-02-24

**Recommendation:** Accept with minor revision

**Comment:**

The reviewers recommended the following suggestions for revision:

- an extended discussion of related works on multi-agent bandits
- provide more real-world motivations on the three settings studied in the paper
- explain why removing the exponential factor in the current regret bounds may be challenging
- clarifications and typos (as also acknowledged in the author feedback)


I also had the same confusion as Reviewer KpK6 even after reading the rebuttal: since linear bandits generalizes multi-arm bandits (with theta being the reward vector), why is the stated problem easier than the bandit case? More explanations will be appreciated.

**Audience:**

Yes

**Claims And Evidence:**

Yes

---

> ### Author Response · Authors · 2025-03-11
> **Camera Ready Version Submitted**
>
> Thank you editors and reviewers for all the constructive comments. We have implemented the changes to the best of our ability and submitted a camera-ready version of our paper.